# Long-Term Effects of Ambient Particulate and Gaseous Pollutants on Serum High-Sensitivity C-Reactive Protein Levels: A Cross-Sectional Study Using KoGES-HEXA Data

**DOI:** 10.3390/ijerph191811585

**Published:** 2022-09-14

**Authors:** Ji Hyun Kim, Hae Dong Woo, Sunho Choi, Dae Sub Song, Jung Hyun Lee, Kyoungho Lee

**Affiliations:** Division of Population Health Research, Department of Precision Medicine, Korea National Institute of Health, Korea Disease Control and Prevention Agency, 200 Osongsaengmyeong 2-ro, Osong-eup, Heungdeok-gu, Cheongju-si 28160, Korea

**Keywords:** ambient air pollution, particulate matter, gaseous pollutant, long-term exposure, high-sensitivity C-reactive protein, systemic low-grade inflammation, cross-sectional study

## Abstract

Ambient air pollutants reportedly increase inflammatory responses associated with multiple chronic diseases. We investigated the effects of long-term exposure to ambient air pollution on high-sensitivity C-reactive protein (hs-CRP) using data from 60,581 participants enrolled in the Korean Genome and Epidemiology Study-Health Examinees Study between 2012 and 2017. Community Multiscale Air Quality System with surface data assimilation was used to estimate the participants’ exposure to criteria air pollutants based on geocoded residential addresses. Long-term exposure was defined as the 2-year moving average concentrations of PM_10_, PM_2.5_, SO_2_, NO_2_, and O_3_. Multivariable linear and logistic regression models were utilized to estimate the percent changes in hs-CRP and odds ratios of systemic low-grade inflammation (hs-CRP > 3 mg/L) per interquartile range increment in air pollutants. We identified positive associations between hs-CRP and PM_10_ (% changes: 3.75 [95% CI 2.68, 4.82]), PM_2.5_ (3.68, [2.57, 4.81]), SO_2_ (1.79, [1.10, 2.48]), and NO_2_ (3.31, [2.12, 4.52]), while negative association was demonstrated for O_3_ (−3.81, [−4.96, −2.65]). Elevated risks of low-grade inflammation were associated with PM_10_ (odds ratio: 1.07 [95% CI 1.01, 1.13]), PM_2.5_ (1.08 [1.02, 1.14]), and SO_2_ (1.05 [1.01, 1.08]). The odds ratios reported indicated that the exposures might be risk factors for inflammatory conditions; however, they did not reflect strong associations. Our findings suggest that exposure to air pollutants may play a role in the inflammation process.

## 1. Introduction

Air pollution is a complex mixture of particulate and gaseous materials that vary depending on their source and physicochemical composition [1,2]. Ambient air pollution has become a major public health burden worldwide, contributing to increased morbidity and premature death, and loss of disability-adjusted life-years [3,4]. Air pollutants have detrimental effects on human health, resulting in multiple chronic diseases including respiratory, cardiovascular, ischemic stroke, and other neuro-metabolic dysfunctions [5,6,7,8]. The underlying mechanisms inducing adverse health outcomes by inhaled air pollutants have not been clearly elucidated, but it is hypothesized that pollutants may activate cellular signaling networks, leading to systemic inflammatory responses [9,10].

Inflammasomes activated by air pollutants induce increased secretion of pro-inflammatory cytokines (e.g., interleukin-6 and tumor necrosis factor α [TNF-α]), and further mediate the hepatic production of acute-phase proteins, including C-reactive protein (CRP) and fibrinogen [11,12]. Among the acute-phase reactants, CRP is a robust marker of low-grade inflammation and is one of the most intensively studied biomarkers associated with cardiovascular health [13,14]. The assessment of high-sensitivity CRP (hs-CRP) is more precise than the conventional measurements, and improved sensitivity makes hs-CRP suitable for detecting low-grade inflammation and the risk of cardiometabolic disorders in the general population [15,16].

Many studies have reported the short-term exposure effects of air pollutants on CRP [17,18]. Increased systemic levels of CRP have been reported after a delay of two days following pollutant exposure because the liver requires time to process acute-phase proteins [19,20]. A recent meta-analysis comprising 40 observational studies with 244,681 participants reported on the association of particulate matters (PMs; particulate matter with aerodynamic diameter < 10 µm [PM_10_] and <2.5 µm [PM_2.5_]) and circulating CRP level. In this study, long-term exposure to particulate air pollution was reported to have a stronger effect as compared to short-term exposure on CRP levels. This indicates substantial cumulative effects of tissue damage and inflammation over longer durations [17]. Moreover, a meta-analysis that included 27 studies on the associations between gaseous pollutants (sulfur dioxide [SO_2_], nitrogen dioxide [NO_2_], and ozone [O_3_]) and CRP levels have yielded inconclusive results and reported that there was a limited number of studies, especially with regards to long-term exposure [18].

In the context of inflammatory markers, PMs have been frequently investigated [14], while prior studies on gaseous materials are limited, and further studies are required [18]. Aggregated long-term concentrations of gaseous pollutants based on daily concentrations of NO_2_, SO_2_, and O_3_ have been suggested as suitable indicators of air pollution, as they often share the same sources, and their levels tend to be highly correlated over time [21]. Various sources of air pollutant measures have become accessible, and the air quality prediction model enabled spatially detailed exposure in domestic areas, including places where the monitoring sites are sparse [22,23]. Moreover, a previous study in a single capital city in Korea assessed the inflammatory effect of a series of air pollutants and suggested that further studies are required to generalize the results obtained to other domestic regions [24].

Accordingly, the current study aimed to fill these gaps by examining the associations between long-term exposure to air pollutants, including PM_10_, PM_2.5_, SO_2_, NO_2_, and O_3_, with the circulating concentrations of hs-CRP in participants from domestic regions.

## 2. Materials and Methods

### 2.1. Study Population

This cross-sectional study was based on the data from the first phase of the follow-up examination (2012–2017) of the Health Examinee (HEXA), a nationwide population-based cohort among the Korean Genome and Epidemiology Study (KoGES).

The study design has been described in detail previously [25]. Briefly, the inclusion criteria were: participants enrolled in KoGES-HEXA; aged over 40 years at baseline; and who visited the health examination centers and hospitals located in metropolitan areas and major cities. Participants underwent a series of anthropometric measurements and clinical examinations. A self-administered questionnaire including socio-demographic information, lifestyle factors, and medical history of diseases was also obtained.

Written informed consent was obtained from all participants prior to the examinations, and the study protocol was approved by the Institutional Review Boards of the National Institute of Health, Korea (2019-05-04-2C-A; 2022-04-04-P-A).

### 2.2. Participant Selection

In the first phase of follow-up between 2012 and 2017, 65,616 participants residing in 16 administrative divisions of South Korea (seven metropolitan cities and nine provinces) were enrolled. Those with incomplete information on residential address, or participants residing in Jeju, the province that differs from other administrative divisions environmentally [26], and participants without linkage with air pollutants data were excluded from the study (2601 observations). We also excluded participants with missing values of hs-CRP and levels higher than 10 mg/L indicating potential acute infections [13] (768 observations). We further excluded participants with missing values in any of the covariates including blood lipid profiles and fasting glucose, blood pressure, body mass index, health-related behaviors, socio-demographic factors, and self-reported physician-diagnosed diseases (1666 observations). The final sample of the study comprised 60,581 participants (Figure 1).

### 2.3. Ambient Air Pollution and Meteorological Factors

Geographic estimation of air pollution and meteorological factors

To estimate the ambient air pollutants and meteorological parameters, first, the concentrations of criteria air pollutants (PM_10_, PM_2.5_, SO_2_, NO_2_, and O_3_) were calculated in a spatial resolution of 9 km using the Community Multiscale Air Quality (CMAQ) model based on the meteorological and emission models. The meteorological values were generated using the Weather Research and Forecasting model (WRF) and used as the input for the emission and chemical transport models and as final meteorological variables for the analysis. The detailed configurations for the meteorological, emission and CMAQ models in this study are presented in Appendix A. Second, surface data assimilation, a method of combining the estimated values with the observed values was applied by assigning a weight to each observation within the radius of influence. The ambient air quality monitoring stations in China and Korea were used in the model. Pun’s interpolation was used for assimilation to produce realistic gradients with sparse air quality data. Third, the spatial resolution of PM_10_ and PM_2.5_ were improved from 9 km to 1 km grid unit by using the aerosol optical depth (AOD) observations from the National Aeronautics and Space Administration Terra and Aqua satellites. Fourth, PM_10_, PM_2.5_, and O_3_ were adjusted for normalized difference vegetation index and meteorological data using the multiple linear regression method. Additionally, to improve the probable overestimation of PM_2.5_ concentrations during the periods when the air quality monitoring data for PM_2.5_ were unavailable (before 2015), the ratio of PM_2.5_ to PM_10_ for dust storm days and several districts of Seoul with PM_2.5_ monitoring data were further extrapolated to all of the regions.

The final daily average levels of meteorological data (e.g., temperature [°C] and humidity [%]) and gaseous pollutants (SO_2_, NO_2_, and O_3_) were presented in every 9 km-grid; and PMs are provided as 1 km-grid unit. All the air pollution and meteorological data were generated as a geocode. The modeling grid points where the air quality monitoring stations are located were extracted; and the performances of modeled concentrations of criteria air pollutants were tested against the measured data from the monitoring stations in Korea. The results of the model performance evaluation for 2010–2017 are provided in Appendix A. The average R^2^ for PM_10_, PM_2.5_, SO_2_, NO_2_, and O_3_ were 0.76, 0.66, 0.42, 0.74, and 0.66, respectively. A more detailed methodology and descriptions for the models have been described in the previous studies [22,27].

Geocoding of the study participants

Participants’ residential addresses were converted into latitude and longitude using a geocoding software, GeoService-Xr (GeoService, Seoul, Korea). If the address for the first phase of follow-up was missing, the address from the baseline survey was used as a substitute. When the detailed address was unavailable or out of date, the address was replaced with the nearby public administrative institution.

Merging the estimated air pollution and meteorological data with the participants

The geocoded addresses of participants were matched to air pollutant and meteorological data with corresponding grid units using the ArcGIS program (ESRI, Inc., Redland, CA, USA). To estimate the individual exposure of each participant, the data from the KoGES and air pollution and meteorological factor database were combined based on the participants’ date of examination and their geocoded units of residential address.

Long-term exposures to ambient air pollutants were defined as the 2-year moving average (lag 0 to 729 days) and short-term exposures were defined as the 3-day moving average (lag 0 to 2 days). To minimize the exposure misclassification errors due to the residential relocation of study participants between baseline data collection and follow-up, we considered the residential mobility during the first visit and follow-up, by using time-weighted average concentrations [28,29]. We assumed that the participants change their place of residence at the midpoint between baseline and the follow-up examination. For instance, if a person had reported residing in “place A” at the baseline exam (1 January 2007) and “place B” at the first follow-up exam (31 December 2008), then the air pollutant exposure in the first half period (from 1 January 2007 to 31 December 2007) was calculated based on place A, and second half period (from 1 January 2008 to 31 December 2008) was based on place B.

### 2.4. Inflammatory Marker

Venous blood samples were drawn after a minimum of 8 h overnight fasting for biochemical analysis, and samples were immediately separated by centrifugation and then sent to a central laboratory authenticated via external quality assessment. The serum hs-CRP levels were quantified by turbidimetric immunoassay method using an automated analyzer (Roche/Hitachi Modular P800; Roche Diagnostics, Tokyo, Japan), with a detection threshold of 0.01 mg/L. Hs-CRP was dichotomized at the level of 3 mg/L, a cutoff point commonly used as a clinical indicator for systemic low-dose inflammation and high-risk cardiovascular disease as well [13].

### 2.5. Covariates

The individual-level covariates were selected a priori and included in the main model following a variance inflation factor (VIF) test with a cut-off value of <5.0 to avoid multi-collinearity. The continuous covariates included were age (years), body mass index (BMI, kg/m^2^), and 2-day moving average of meteorological factors (temperature [°C] and relative humidity [%]) and the categorical covariates included were sex, visit year, season, weekday of examination, place of residence (metropolitan and non-metropolitan) [30], smoking status (non, ex, and current), drinking status (non, ex, and current), regular exercise (yes and no to whether the study participants regularly exercise to make their bodies sweat at least once a week), occupation (professional, administrative; office, sales, and service; laborer, agricultural; and others, unemployed), education (less than middle school [9 years], middle-high school [9–12 years], and college or more [>12 years]), marital status (married/cohabiting; and single/divorced/widowed/separation, or others), medical history of diabetes (defined as self-reported physician-diagnosed diabetes, or fasting blood glucose ≥ 126 mg/dL), hypertension (defined as self-reported physician-diagnosed hypertension, or systolic blood pressure ≥ 140 mmHg, or diastolic blood pressure ≥ 90 mmHg), dyslipidemia (defined as self-reported physician-diagnosed dyslipidemia, or triglyceride ≥ 200 mg/dL, or total cholesterol ≥ 240 mg/dL, or low-density lipoprotein-cholesterol ≥ 160 mg/dL, or high-density lipoprotein-cholesterol < 40 mg/dL) [31], cardio- and cerebrovascular disease (defined as self-reported physician diagnosis of ischemic heart disease or cerebral stroke), and cancer (defined as self-reported physician diagnosis of any types of cancers).

### 2.6. Statistical Analyses

For multivariable linear models, hs-CRP was log_e_-transformed before the analysis to better approximate normality. We assessed the association of long-term exposure to ambient air pollutants with the level of log_e_-transformed hs-CRP. The pollutant effects were expressed as (exp(β) − 1) ∗ 100, where β was the estimated regression coefficient, and the results of the analyses were presented as a percentage difference of hs-CRP per interquartile range (IQR) increase in air pollutants. For multivariable logistic regression models, we examined the probability of hs-CRP > 3 mg/L [13].

Sensitivity analyses were performed to evaluate the robustness of our findings. We evaluated three models, each with incrementally more covariates: model 1 included age, sex, time, and meteorological factors; model 2 included the covariates in model 1 as well as for lifestyle factors; and model 3 (main model) included the covariates in model 2 as well as for the medical history of diseases. We conducted subgroup analyses by age (<65 or ≥65 years), sex (male or female), BMI (<25 or ≥25 kg/m^2^; obese was defined as a BMI of 25 kg/m^2^ or higher based on the Asian-Pacific criteria [32]), smoking status (non-, ex-, or current smokers), regular exercise (yes or no), and self-reported medical history of diabetes, hypertension, dyslipidemia, and cardio- and cerebrovascular disease (yes or no) for multivariable-adjusted linear and logistic regression models, and subgroup analysis by drinking status (non-, ex-, or current drinkers) was also carried out for logistic models. The effect modification was evaluated by including multiplicative terms between each pollutant and potential effect modifier in the adjusted models. We also examined the association after adjusting for short-term exposure, defined as the 3-day average of each pollutant, with and without interaction terms between long- and short-term exposure of each pollutant, to further consider whether the chronic effect was attenuated by the acute effect [24]; and investigated the remaining predictive effects of pollutants by incorporating two-pollutant models [33], with the restriction of Spearman inter-correlation between pollutants below 0.70 [14]. Furthermore, to examine whether there is a potential misclassification in exposure, we conducted a subgroup analysis by only including participants who did not change their residential address between the baseline and the first phase of follow-up.

All statistical analyses were performed using SAS software version 9.4, Cary, NC, USA. Two-tailed *p*-values < 0.05 were considered significant except for the correlation analyses with a more conservative level of significance (*p* < 0.001), and for interaction terms where we accepted *p* < 0.10.

## 3. Results

### 3.1. Study Population

The characteristics of the participants are presented in Table 1. The mean age of the participants was 58.7 ± 8.1 years, and 66.5% of them were women. Most of the participants resided in metropolitan areas (62.5%), were non- or ex-smokers (92.3%), were non- or ex-drinkers (58.9%), were non-obese (67.7%), and exercised regularly (58.8%). Most of the participants had no history of diabetes (86.5%), cardio- and cerebrovascular disease (94.4%), cancer (94.0%), or lung diseases (asthma [97.8%]; and chronic obstructive pulmonary disease [99.8%]). When comparing the two groups according to the low-grade inflammation status, a higher proportion of participants with inflammation (hs-CRP > 3 mg/L) were above 65 years of age, men, obese, those who exercised less regularly, those with a history of diseases, and those dwelling in a non-metropolitan area as compared to participants with normal hs-CRP levels (hs-CRP ≤ 3 mg/L). Additionally, a lower proportion of participants with inflammation were non-smokers and non-drinkers as compared to participants with normal hs-CRP levels. The groups also differed in occupation distribution, education attainment, and season of enrollment.

### 3.2. Air Pollutants

The summary statistics of environmental factors including 2-year average exposure measures of air pollutants and a 2-day average of meteorological factors are summarized in Table 2 and Appendix A. The participants’ average exposure to PM_10_, PM_2.5_, SO_2_, NO_2_, and O_3_ were 48.34 μg/m^3^, 25.20 μg/m^3^, 5.40 ppb, 23.46 ppb, and 25.86 ppb, respectively. The exposures in all participants exceeded the annual World Health Organization (WHO) air quality guideline (AQG) levels for PMs, which has been updated in 2021 (PM_10_ < 15 μg/m^3^; and PM_2.5_ < 5 μg/m^3^) [34]. As per the Korean AQG for annual average limits (PM_10_ < 50 μg/m^3^; and PM_2.5_ < 15 μg/m^3^) [35], 20,328 (33.56%) and 60,567 (99.98%) participants exceeded the limits of PM_10_ and PM_2.5_, respectively. With regards to annual WHO AQG levels for NO_2_ (<10 μg/m^3^) [34], 36,397 (60.08%) participants exceeded the limit. As per the Korean AQG for annual average limits (SO_2_ < 20 ppb; and NO_2_ < 30 ppb), 7 (0.01%) and 15,435 (25.48%) participants exceeded SO_2_ and NO_2_ levels, respectively [35]. No annual guidelines or limits have been established for O_3_.

The strongest correlation was observed between NO_2_ and O_3_ (r_s_ = −0.78), followed by PM_10_ and O_3_ (r_s_ = −0.66), NO_2_ and SO_2_ (r_s_ = 0.62), PM_10_ and PM_2.5_ (r_s_ = 0.59), and NO_2_ and PM_10_ (r_s_ = 0.50). Temperature exhibited no significant correlation with PMs.

### 3.3. Effects of Long-Term Exposure to Air Pollutants on Inflammatory Markers

The results of the associations of long-term residential exposure to ambient air pollutants with hs-CRP level are presented in Figure 2. Significant concentration-response associations were observed in all pollutants. This was evident as participants with higher-quartile exposure generally had higher levels of hs-CRP in PM_10_, PM_2.5_, SO_2_, and NO_2_, while a negative trend in the association was observed in O_3_ (all *p* < 0.001). Each IQR increase in 2-year average concentrations of air pollutants was significantly associated with percent changes in hs-CRP levels after controlling for potential confounders (PM_10_: 3.75%, 95% confidence interval (CI) 2.68, 4.82; PM_2.5_: 3.68%, 95% CI 2.57, 4.81; SO_2_: 1.79%, 95% CI 1.10, 2.48; NO_2_: 3.31%, 95% CI 2.12, 4.52; and O_3_: −3.81%, 95% CI −4.96, −2.65).

Figure 3 shows the odds ratios (ORs) and 95% Cis for the association between long-term exposure to ambient air pollutants with low-grade inflammation (level of hs-CRP > 3 mg/L). Positive trends in the associations were observed between 2-year average concentrations of PM_10_, PM_2.5_, and SO_2_ with hs-CRP levels, indicated by an increment in the magnitude of point estimates (all *p* < 0.05). Each IQR increase in 2-year average concentrations of PMs and SO_2_ was significantly associated with elevated risks of low-grade inflammation in the fully adjusted model (PM_10_: 1.07, 95% CI 1.01, 1.13; PM_2.5_: 1.08, 95% CI 1.02, 1.14; and SO_2_: 1.05, 95% CI 1.01, 1.08).

### 3.4. Sensitivity Analyses

When we evaluated the models, each with incrementally more covariates, the direction, strength, and significance of associations were maintained from the minimally adjusted model (model 1) to the fully adjusted model (model 3; main model) (Appendix A).

The effect modifications on the association between each air pollutant and hs-CRP are presented in Appendix A for linear regression models and Appendix A for logistic regression models. In our linear models, statistically significant effect modifications were observed with age (PM_2.5_), sex (PM_10_, NO_2_, and O_3_), BMI (PM_10_, PM_2.5_, and SO_2_), smoking status (SO_2_ and NO_2_), regular exercise (SO_2_ and NO_2_), and medical histories of hypertension (PM_10_ and PM_2.5_), dyslipidemia (PM_10_), and cardio- and cerebrovascular disease (SO_2_ and NO_2_) (all *p*-values < 0.10). In the logistic regression, similar effect modifying patterns were observed for smoking status (SO_2_ and NO_2_), regular exercise (SO_2_), and histories of hypertension (PM_10_, NO_2_, and O_3_) and cardio- and cerebrovascular disease (SO_2_ and NO_2_) (all *p*-values < 0.10). However, no clear effect modifying patterns were observed across all pollutants.

The summary statistics of short-term exposure to air pollutants and Spearman’s correlation coefficients among short- and long-term metrics are described in Appendix A. The models simultaneously adjusting for the short-term exposure measures without interaction terms remained stable or demonstrated slightly attenuated effect estimates (Appendix A). When additionally adjusting for the multiplicative interaction term between long- and short-term exposure, it appeared that the effect estimates in our linear models were stronger for SO_2_ and NO_2_, whereas O_3_ demonstrated an attenuated effect and was no longer statistically significant. The significant association between SO_2_ and low-grade inflammation identified in the main model changed to non-significant in models with and without the interaction term. In two-pollutant models, some associations observed with single pollutants were attenuated after adjusting for a second pollutant (Appendix A). The combined effect estimates for PMs were attenuated after adjusting for each other. Similarly, the estimates for gaseous pollutants (SO_2_, NO_2_, O_3_) weakened substantially after adding PM_10_ to the model.

We also considered another sensitivity analysis for participants who did not change their residential address between baseline and the first phase of follow-up (Appendix A). When the level of hs-CRP was treated as a continuous or dichotomous variable, the estimated effects remained stable, and the direction of the association did not differ largely from the main model. The odds of low-grade inflammation became non-significant in PM_2.5_, while in the case of O_3,_ a marginally reduced risk of low-grade inflammation (OR 0.71, 95% CI 0.51, 1.00) was observed.

## 4. Discussion

### 4.1. Summary

This study investigated the association between long-term exposure to air pollutants and hs-CRP level. When the hs-CRP concentration was treated as a continuous variable, statistically significant and positive associations were observed for PM_10_, PM_2.5_, SO_2_, and NO_2_, while a negative association was found for O_3_. Further, an increased risk of low-grade inflammation (hs-CRP > 3 mg/L) was observed with PM_10_, PM_2.5_, and SO_2_. The odds ratios reported indicated that the exposures might be risk factors for inflammatory conditions; however, they did not reflect strong associations.

### 4.2. Comparisons of Findings

In this study, significant positive increases were observed in the 2-year average PMs, and the magnitude of association was larger for PM_2.5_ compared to PM_10_: 10 μg/m^3^ increase in PM_10_ and PM_2.5_ were associated with 6.38% (95% CI 4.55 to 8.24) and 10.27% (95% CI 7.09 to 13.54) increase in hs-CRP, respectively. PM_2.5_ exposure may be more hazardous to human health compared to PM_10_, as PM_2.5_ mainly originates from the secondary aerosol sources via a chemical reaction of gaseous pollutants (e.g., SO_2_ and NO_2_), whereas the constituents of PM_10_ are mainly contributed by soil dust and natural sources [36]. PM_10_ can also be deposited majorly in the large conducting airways, while PM_2.5_ can penetrate and reach the alveolar-capillary barrier, and then travel to other organs within the body; and therefore, PM_2.5_ may have a higher potential for inducing inflammation and oxidative stress [6,9].

Regarding the long-term exposure to PMs and the level of CRP, a meta-analysis with a pooled estimate of five and nine studies of long-term exposure to PM_10_ and PM_2.5_ indicated a significant 5.61% and 18.01% increase in CRP per 10 μg/m^3^ increase, respectively [17]. Several observational studies that examined both PM_10_ and PM_2.5_ have reported a greater impact of PM_2.5_ compared to PM_10_ [37,38,39]. A panel study of the capital city in Korea also demonstrated a slightly higher effect estimate in the association between CRP and PMs; per 10 μg/m^3^ increase in the annual average concentrations of PMs with 0.34 and 0.47 percent changes of CRP for PM_10_ and PM_2.5_, respectively [24]. The larger magnitude of the association of PMs with CRP in our study compared to that observed in a previous Korean study [24] may be partially explained by the exposure assessment methods, as the effect size of the subgroup of fixed-site monitoring studies tended to be lower than studies which used other measurement methods (e.g., modeled data or personal monitoring) [17].

Among gaseous pollutants, significant positive associations were observed with the hs-CRP levels in SO_2_ and NO_2_, while negative associations were observed in O_3_. Adjustment for ambient PM_10_ substantially attenuated the associations with SO_2_ and NO_2_ in both continuous and binary measures of hs-CRP, given that particles would partly capture the complex combustion-related constituents [40]. This conforms with the source apportionment of PMs in Korea that explained the secondary aerosol sources via the atmospheric chemical reaction of SO_2_ and NO_2_ originated from fuel combustion [36]. Accordingly, the significant effects of gaseous pollutants on hs-CRP in this study might have been due to the shared emission sources, with significantly correlated particulate and gaseous pollutants [21].

Several studies that measured multiple pollutants have also reported similar findings. Cross-sectional studies representative of the English population demonstrated positive but non-significant combined estimates for SO_2_ and NO_2_ with CRP, while the estimate for O_3_ was in the opposite direction [41]. Moreover, another cross-sectional study of the Chinese population reported that the annual ambient SO_2_ and NO_2_ concentrations were positively associated with low-grade inflammation (statistically significant for NO_2_ and non-significant for SO_2_), whereas O_3_ exposure exhibited a significant inverse association [42]. However, a recent meta-analysis on the association between gaseous air pollutants and CRP has reported conflicting results across studies; and in both short- and long-term gaseous pollutants and CRP levels, positive non-significant pooled estimates were observed for SO_2_, NO_2_, and O_3_ as well [18]. A review on the health effects of O_3_ suggested that in contrast to the well-established evidence from short-term studies, the evidence for the chronic effects is less conclusive and lacks research; therefore, further investigations are needed to better describe the relationship between O_3_ and chronic health endpoints [43].

Unlike other pollutants, long-term exposure to O_3_ demonstrated a decreased risk of hs-CRP and systemic low-grade inflammation in this study. There are uncertainties in interpreting this seemingly protective association, but several plausible explanations may be offered. First, as protective effects are not expected according to the literature and biological plausibility, the seemingly beneficial impact of O_3_ on the inflammatory response might have been artifacts of pollutants that were positively correlated with other factors and were negatively correlated with CRP (e.g., O_3_-derivatives) [44]. Second, O_3_ may have a threshold level for its effect on human health. A cross-sectional study in Germany identified a J-shaped association between short-term exposure to O_3_ and hs-CRP levels [45]. In particular, for participants with O_3_ concentration less than 120 µg/m^3^, the level of hs-CRP was negatively associated with O_3_ [45]. All participants in our study were exposed to O_3_ concentration far lower than 120 µg/m^3^. However, direct comparisons of the results should be cautiously interpreted due to possible differences between the mechanisms underlying short- and long-term exposure to O_3_ [42]. Lastly, O_3_ may actually show an inverse association with multiple metabolic abnormalities, including other markers of inflammation or coagulation, blood lipid profiles, and glucose homeostasis. A cross-sectional study in China found that long-term exposure to ambient NO_2_ and O_3_ were risk and protective factors of low-grade systemic inflammation, respectively; and also demonstrated that O_3_ was negatively associated with multiple metabolic indicators of lipid metabolism and glucose homeostasis, while NO_2_ was positively associated with them [42]. We also performed additional analyses and found that the blood levels of white blood cells, fibrinogen, triglycerides, and fasting blood glucose were positively associated with PMs, SO_2_, and NO_2_, and inversely associated with O_3_ (Appendix A).

### 4.3. Sensitivity Analyses on Effect Modifications

Along with variations in the concentrations of air pollutants, the susceptibility of a population also plays a crucial role in inflammatory responses [14]. In this study, elderly participants, women, obese participants, sedentary participants, current smokers, and participants with comorbidities were potentially susceptible to air pollutants. However, there were no clear effect modifying patterns across all air pollutants.

Elderly individuals are generally considered more susceptible to the adverse health impacts of ambient air pollution [46]. Aging entails a substantial burden regarding the onset and exacerbation of chronic disorders, perturbations in immune system function, and changes in activity status [47,48]. These findings are consistent with the associations between pollutants and CRP reported by several observational studies of susceptible older individuals, including factors such as diabetes, obesity, hypertension, coronary heart disease, and chronic obstructive pulmonary disease [49,50,51,52]. Sex-based differences in immune and inflammatory responses may also partly explain the greater susceptibility to air pollutants observed in women. Compared to men, women may have a higher CD4^+^:CD8^+^ lymphocyte ratio; and when stimulated, their blood monocytes generate more prostaglandin E2 and less TNF-α [53].

Participants who were ex-smokers appeared to be less vulnerable to air pollutants, while current- or non-smokers exhibited greater vulnerability. Similar trends were reported in a cross-sectional study in Germany; however, the study stated that the generalizability of their results might have been limited because smokers tended to be younger and healthier (lower percentages of diabetes or cardiovascular disease) than were ex- or non-smokers [14]. When we also stratified the participants by smoking status, current smokers were younger and healthier (lower percentages of diabetes, hypertension, cardio- and cerebrovascular disease, cancer, or lung diseases) compared to ex-smokers, suggesting that differences in the participants’ characteristics may be the reason (Appendix A). However, the potential modifying role of smoking on the air pollution-associated inflammatory response remains unclear. One hypothesis suggested that smoking itself may trigger anatomical damage, which may exert additional harmful effects and exacerbate the adverse effects of ambient air pollution [54,55,56]. Studies supportive of this perspective have suggested that inflammatory responses to air pollutants appeared to be more pronounced in smokers [52,57,58,59]. A controversial hypothesis explained that smokers may have elevated levels of oxidative stress and inflammatory response, and relatively small variations in levels of inflammatory markers could be attributed to ambient air pollution; on the other hand, non-smokers may have minimal influence from smoking-induced inflammation [60]. Studies supportive of this perspective have suggested that inflammatory responses to air pollutants appeared to be more pronounced in non-smokers [38,60,61].

Moreover, participants who were not maintaining regular physical activity or were obese appeared to be more vulnerable to air pollutants in this study. Similar results were reported by a Taiwanese cohort study; compared with inactive participants with PM_2.5_ in the highest quartile, participants in other groups had lower white blood cell counts, especially among those with higher levels of physical activity and lower exposure to PM_2.5_ [62]. A higher frequency of moderate-to-vigorous physical activity was significantly associated with a lower risk of cardiovascular diseases in the groups of both high and low PMs as well [63]. Except for the extremely polluted environments, the benefits of physical activity may outweigh the health risks caused by air pollution [64]. According to a systematic review of air pollution on obesity, the probable biological pathways in the included studies involved enhanced oxidative stress and inflammatory responses of adipose tissues, elevated risk of chronic comorbidities, and insufficient physical activity; however, the impact of pollution on the body weight status remained unclear [65].

Cumulatively, the evidence on the extent to which individual characteristics modify the effect of air pollution through inflammation remains controversial across studies. The results should be interpreted cautiously due to complexities including geographic regions, study design, selection of the subpopulation, and types, sources, composition, toxicity, or dose of pollutants. However, it should be noted that the global health burden of air pollution and lifestyle habits co-exists and the annual level of PM among participants exceeds the annual AQG provided by the WHO [34]. Therefore, public health interventions to maintain healthier lifestyles may potentially minimize the detrimental health impacts due to ambient air pollutants (e.g., elevated inflammatory responses and increased risk of multiple chronic diseases) and should be incorporated into air pollution reduction strategies.

### 4.4. Sensitivity Analyses on Extended Models including Short-Term Measures and Co-Pollutants

To examine the attenuation of the chronic effects of air pollutants by their acute effects, we simultaneously adjusted for short-term exposure measures of corresponding pollutants (3-day average) in our models. The results remained stable or demonstrated slightly attenuated effect estimates. Short-term air pollution exposure may potentially affect the long-term associations between inflammatory markers and outdoor air pollution exposure; however, participants with potential acute infections (hs-CRP levels > 10 mg/L) were excluded before analysis, which may have reduced the potential confounding effect of short-term exposure to air pollutants on the chronic health effects of air pollutants [48]. Previous epidemiological studies of long-term exposure to air pollutants on hs-CRP levels also included participants with hs-CRP levels less than 10 mg/L and demonstrated stable estimates which were not affected by adjusting for short-term exposure to air pollution [38,66]. A previous study in a single capital city in Korea with participants with CRP levels ranging from 0.1 to 11.7 mg/L did not report any significant changes after adjusting for short-term exposure, whereas the white blood cell counts, another clinical marker of inflammation and a potential predictor of cardiovascular risk, demonstrated significant changes in the long-term association with air pollution after adjustment; further investigations are suggested to confirm these findings [24].

As humans are exposed to polluted air containing a complex mixture of particle and gaseous pollutants rather than an isolated single pollutant, multi-pollutant concepts should also be considered [67,68]. Therefore, to evaluate the remaining predictive effects of air pollutants, we simultaneously adjusted for other co-pollutants by adding one pollutant at each time in our two-pollutant models. The effect estimates for PMs were attenuated after adjusting for each other and the effect estimates of PM_10_ decreased from 6.38 to 5.69 while PM_2.5_ estimates decreased from 10.27 to 1.73 (units: per 10 µg/m^3^ increase). Similar two-pollutant associations were also observed in the previous cross-sectional study with PMs controlled for each other; the effect estimates of PM_10_ increased slightly from 5.15 to 7.34 while PM_2.5_ estimates decreased largely from 0.16 to −5.61 (units: per IQR increase) [14]. This phenomenon suggests non-independent effects of PM_10_ and PM_2.5_ on hs-CRP and may strengthen the importance of PM_10_ based on the substantial heterogeneity of exposure levels across the study regions compared to PM_2.5_ [14]. We also observed that the effect estimates for gaseous pollutants (SO_2_, NO_2_, O_3_) weakened substantially after adding PM_10_ to the model: the effect estimates were largely attenuated from 30.77 to 16.81, 5.22 to 1.74, and −16.13 to −7.74 for SO_2_, NO_2_, and O_3_, respectively (units: per 10 µg/m^3^ increase). Compared to PM_10_, adjustment for PM_2.5_ attenuated the effect estimates of gaseous pollutants to a lesser extent and the estimates changed to 24.94, 5.69, and −15.46 for SO_2_, NO_2_, and O_3_, respectively (units: per 10 µg/m^3^ increase). The greater change of effect estimates in the two-pollutant models compared to single-pollutant models may possibly be due to the stronger correlation coefficients of gas phase pollutants with PM_10_ than that of PM_2.5_. However, the two-pollutant associations should be analyzed and interpreted cautiously. As a strong correlation within pollutants in the regression models may become highly unstable, we incorporated the restriction of Spearman inter-correlation between pollutants below 0.70 [14,69]. Therefore, the results for the combination of NO_2_ and O_3_ with high correlation (r_s_ = −0.78) were not reported. Investigating multi-pollutant model to air pollutant research is also challenging due to the other factors including complicated formation pathways, diverse classes of measurement errors (e.g., differential errors across pollutants), and complex epidemiological interpretation (e.g., health effects due to ambient gaseous pollutants which may actually be a result of PM exposure) [69,70,71].

### 4.5. Strengths and Limitations

The strengths of this study are as follows. First, the participants exhibited spatial variability, indicated by their diverse domestic geographical residence around the country. Second, diverse individual-level information (e.g., variables of socioeconomic status and lifestyle factors) was available. Third, the ambient air quality prediction model enabled air pollution research not only for the residents in capital or metropolitan areas, but also for those in non-metropolitan areas, where the measurement points are scarce [22]. Especially, the model that incorporated satellite-based reanalysis enabled the estimates of PMs with high resolution, despite the absence of air quality monitoring station data of PM_2.5_ before 2015. Lastly, we attempted to account for residential mobility by considering time-weighted average concentrations of air pollutants. However, the follow-up time between the baseline and follow-up differed for each participant (mean and standard deviation: 4.9 ± 1.7 years). Further linkage with personal records that contain each participant’s annual residential mobility, and the length of exposure needs to be traced [72].

The potential limitations of this study should also be considered. First, due to the cross-sectional design, temporal trends and causal effects of air pollutants on hs-CRP are difficult to determine. Circulating inflammatory markers are in a physiologically varying state; and studies with longitudinal designs or multiple measurements may diminish potential biases more effectively and identify the associations more accurately [73]. Second, potential exposure measurement errors might exist because of the use of the predictions of air pollutants as proxy exposure measurements assigned to the participants’ residences to evaluate the inflammatory effects of pollutants. The exposure misclassification can be increased by the absence of the individuals’ specific activity patterns; however, the exposure data based on indoor combustion sources and occupational contributions were unavailable. Accordingly, both classical and Berkson errors might arise; however, the errors can bias toward the null, suggesting the estimated effect is unlikely to be exaggerated [74,75]. The high-resolution ambient air quality prediction models may result in a smaller bias, which suggests robustness against the exposure error [75]. However, the performance of SO_2_ in our CMAQ models (R^2^ = 0.42) should be discussed. A similar trend was also observed for SO_2_ in the US prediction models which utilized CMAQ with a 12 km grid; even after utilizing the data fusion methods to correct the biases associated with the model output by combining observed data to the CMAQ model, SO_2_ still performed the worst with the lowest R^2^ among the modeled pollutants [76,77]. SO_2_ predictions with CMAQ had the most error due to the challenges of coal combustion plumes [70,76]; and SO_2_ might have a less homogeneous spatial distribution [77]. Further optimization for the emission inventories is needed to improve the prediction accuracy of SO_2_ [78]. Third, our study lacked information on medication. Hs-CRP is largely affected by medication, such as non-steroidal anti-inflammatory drugs or lipid-lowering drugs, including statins [14]. Statins may reduce the associations between long-term pollutant exposure and hs-CRP due to the anti-inflammatory actions of HMG-CoA reductase inhibitors [38]. However, no clear effect modifying patterns with regard to medications were observed in several studies [24,38]. Lastly, residual confounding might have existed in this study. One example is uncontrolled factors with regard to pollutants. In particular, O_3_ concentration is largely affected by local-scale as it is rapidly scavenged by nitric oxide near roadways; thus, considering the roadway proximity is important [42]. However, we could not control for residential distances from main roadways in our models and assess whether the association between ambient O_3_ and hs-CRP remains stable.

## 5. Conclusions

This study investigated the association between long-term exposure to air pollutants and hs-CRP, which is a non-specific marker of inflammation, among participants residing in domestic regions in Korea. Most of the outcomes persisted when hs-CRP was treated as either a continuous variable or as a dichotomous variable using a threshold of 3 mg/L (positive associations for PM_10_, PM_2.5_, SO_2_, and NO_2_, and negative associations for O_3_).

Further well-designed research (e.g., longitudinal, or multiple measurements) with sufficient information on annual residential changes, occupation, indoor exposure, and geographical factors may help elucidate the relationship between inflammatory markers and long-term exposure to gaseous pollutants or chemical constituents involved in the probable mechanistic pathways.

## Figures and Tables

**Figure 1 ijerph-19-11585-f001:**
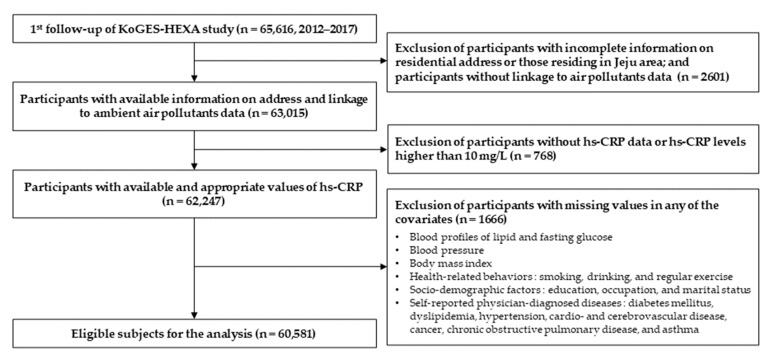
Flow chart of the study participants.

**Figure 2 ijerph-19-11585-f002:**
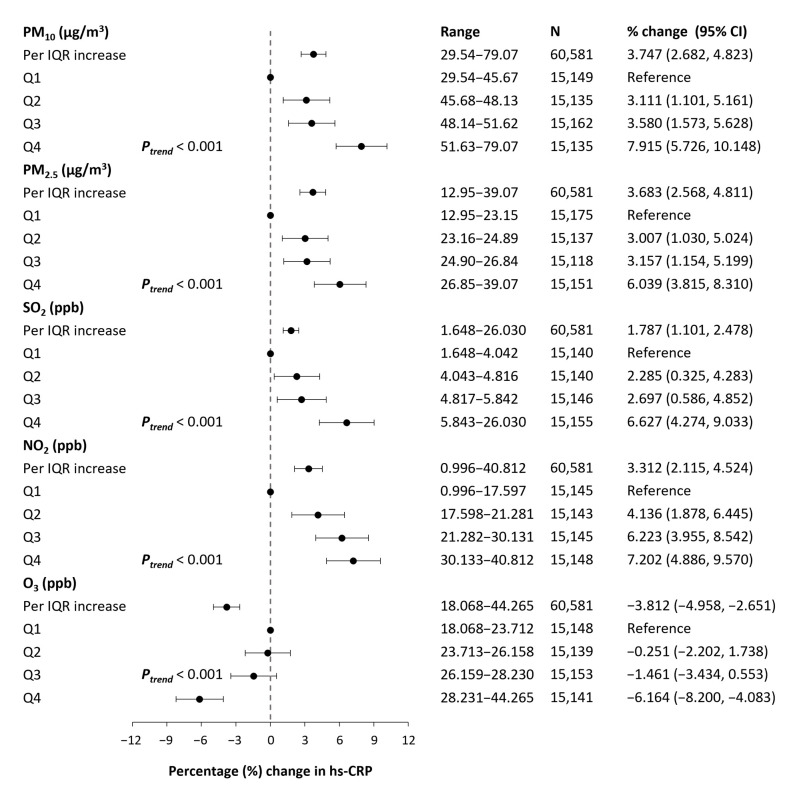
Associations between hs-CRP levels and long-term exposure to air pollutants. The models are adjusted for age, sex, BMI, smoking status, drinking status, regular exercise, occupation, education, marital status, residential area, year, season, and weekday of examination, medical history of diseases (diabetes, hypertension, dyslipidemia, cardio- and cerebrovascular disease, and cancer), and meteorological factors (2-day moving average of temperature and relative humidity). Estimates are presented as percentage changes with 95% confidence intervals in hs-CRP levels per IQR increase in 2-year average ambient air pollution exposure (corresponding IQR of each pollutant: 5.95 μg/m^3^ for PM_10_, 3.70 μg/m^3^ for PM_2.5_, 1.80 ppb for SO_2_, 12.54 ppb for NO_2_, and 4.52 ppb for O_3_). PM_10_, PM_2.5_: particulate matter with aerodynamic diameter < 10 μm and <2.5 μm, respectively; SO_2_: sulfur dioxide; NO_2_: nitrogen dioxide; O_3_: ozone; Hs-CRP: high-sensitivity C-reactive protein; IQR: interquartile range; Q1, Q2, Q3, Q4: first, second, third, and fourth quartile, respectively.

**Figure 3 ijerph-19-11585-f003:**
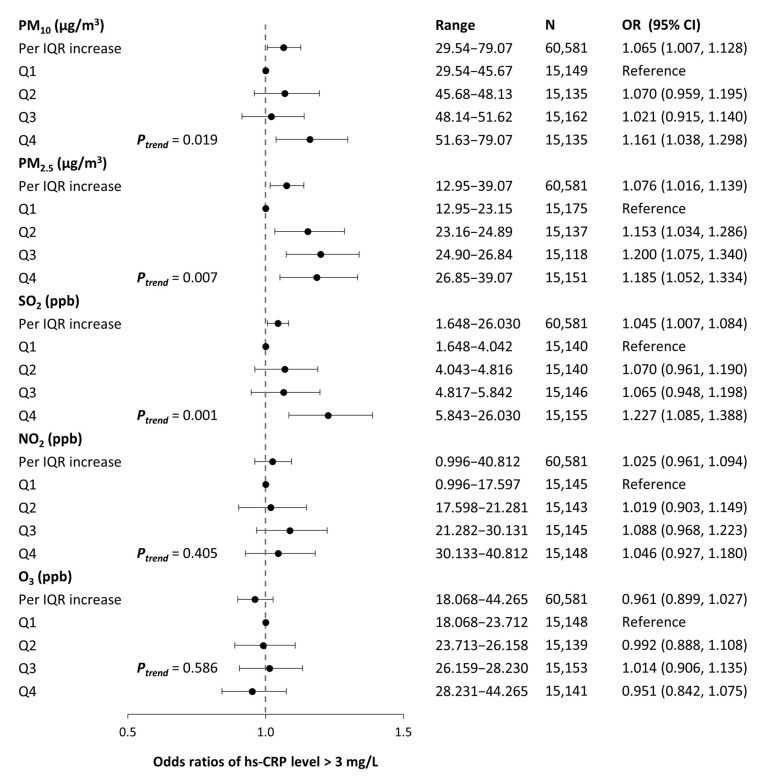
Associations between systemic low-grade inflammation and long-term exposure to air pollutants. The models are adjusted for age, sex, BMI, smoking status, drinking status, regular exercise, occupation, education, marital status, residential area, year, season, and weekday of examination, medical history of diseases (diabetes, hypertension, dyslipidemia, cardio- and cerebrovascular disease, and cancer), and meteorological factors (2-day moving average of temperature and relative humidity). Estimates are reported as the odds ratios and 95% confidence intervals of low-grade inflammation (hs-CRP levels > 3 mg/L) per IQR increase in 2-year average ambient air pollution exposure (corresponding IQR of each pollutant: 5.95 μg/m^3^ for PM_10_, 3.70 μg/m^3^ for PM_2.5_, 1.80 ppb for SO_2_, 12.54 ppb for NO_2_, and 4.52 ppb for O_3_). PM_10_, PM_2.5_: particulate matter with aerodynamic diameter < 10 μm and <2.5 μm, respectively; SO_2_: sulfur dioxide; NO_2_: nitrogen dioxide; O_3_: ozone; OR: odds ratio; IQR: interquartile range; Q1, Q2, Q3, Q4: first, second, third, and fourth quartile, respectively.

**Table 1 ijerph-19-11585-t001:** General characteristics of the study participants.

Characteristics	Total (*n* = 60,581)	Hs-CRP ≤ 3 mg/L (*n* = 57,630)	Hs-CRP > 3 mg/L (*n* = 2951)	*p*-Value ^a^
Age [years (mean ± SD)]	58.7 ± 8.1	58.6 ± 8.1	60.1 ± 8.3	<0.001
Age in categories [*n* (%)]				
41–64 years	44,856 (74.0)	42,855 (74.4)	2001 (67.8)	<0.001
≥65 years	15,725 (26.0)	14,775 (25.6)	950 (32.2)	
Sex [*n* (%)]				
Males	20,282 (33.5)	19,110 (33.2)	1172 (39.7)	<0.001
Females	40,299 (66.5)	38,520 (66.8)	1779 (60.3)	
Body mass index [kg/m^2^ (mean ± SD)]	23.9 ± 2.9	23.8 ± 2.9	24.9 ± 3.4	<0.001
Body mass index in categories [*n* (%)]				
<25 kg/m^2^	40,993 (67.7)	39,370 (68.3)	1623 (55.0)	<0.001
≥25 kg/m^2^	19,588 (32.3)	18,260 (31.7)	1328 (45.0)	
Smoking status [*n* (%)]				
Non-smoker	45,647 (75.4)	43,601 (75.7)	2046 (69.3)	<0.001
Ex-smoker	10,252 (16.9)	9699 (16.8)	553 (18.7)	
Current smoker	4682 (7.7)	4330 (7.5)	352 (11.9)	
Drinking status [*n* (%)]				
Non-drinker	31,544 (52.1)	30,036 (52.1)	1508 (51.1)	0.023
Ex-drinker	4118 (6.8)	3881 (6.7)	237 (8.0)	
Current drinker	24,919 (41.1)	23,713 (41.2)	1206 (40.9)	
Regular exercise [*n* (%)]				
Yes	35,617 (58.8)	34,058 (59.1)	1559 (52.8)	<0.001
No	24,964 (41.2)	23,572 (40.9)	1392 (47.2)	
Occupation [*n* (%)]				
Professional, administrative	6339 (10.5)	6072 (10.5)	267 (9.1)	0.015
Office, sales, and service	13,578 (22.4)	12,946 (22.5)	632 (21.4)	
Laborer, agricultural	9037 (14.9)	8569 (14.9)	468 (15.9)	
Others, unemployed	31,627 (52.2)	30,043 (52.1)	1584 (53.7)	
Education [*n* (%)]				
Less than middle school (<9 years)	9158 (15.1)	8606 (14.9)	552 (18.7)	<0.001
High school (9–12 years)	32,834 (54.2)	31,211 (54.2)	1623 (55.0)	
College or more (>12 years)	18,589 (30.7)	17,813 (30.9)	776 (26.3)	
Marital status [*n* (%)]				
Married, cohabitating	53,778 (88.8)	51,181 (88.8)	2597 (88.0)	0.186
Single, divorced, widowed, separation, others	6803 (11.2)	6449 (11.2)	354 (12.0)	
Medical history				
Diabetes [*n* (%)]				
Yes	8161 (13.5)	7584 (13.2)	577 (19.6)	<0.001
No	52,420 (86.5)	50,046 (86.8)	2374 (80.4)	
Hypertension [*n* (%)]				
Yes	20,982 (34.6)	19,690 (34.2)	1292 (43.8)	<0.001
No	39,599 (65.4)	37,940 (65.8)	1659 (56.2)	
Dyslipidemia [*n* (%)]				
Yes	25,092 (41.4)	23,746 (41.2)	1346 (45.6)	<0.001
No	35,489 (58.6)	33,884 (58.8)	1605 (54.4)	
CCVD [*n* (%)]				
Yes	3390 (5.6)	3183 (5.5)	207 (7.0)	0.001
No	57,191 (94.4)	54,447 (94.5)	2744 (93.0)	
Cancer [*n* (%)]				
Yes	3637 (6.0)	3427 (5.9)	210 (7.1)	0.010
No	56,944 (94.0)	54,203 (94.1)	2741 (92.9)	
Asthma [*n* (%)]				
Yes	1348 (2.2)	1255 (2.2)	93 (3.2)	0.001
No	59,233 (97.8)	56,375 (97.8)	2858 (96.8)	
COPD [*n* (%)]				
Yes	110 (0.2)	98 (0.2)	12 (0.4)	0.007
No	60,471 (99.8)	57,532 (99.8)	2939 (99.6)	
Season [*n* (%)]				
Spring (March–May)	8815 (14.6)	8413 (14.6)	402 (13.6)	<0.001
Summer (June–August)	20,781 (34.3)	19,854 (34.5)	927 (31.4)	
Fall (September–November)	23,693 (39.1)	22,478 (39.0)	1215 (41.2)	
Winter (December–February)	7292 (12.0)	6885 (12.0)	407 (13.8)	
Residential area [*n* (%)]				
Metropolitan	37,860 (62.5)	36,068 (62.6)	1792 (60.7)	0.044
Non-metropolitan	22,721 (37.5)	21,562 (37.4)	1159 (39.3)	
Hs-CRP [mg/L (mean ± SD)]	0.89 ± 1.14	0.68 ±0.57	4.95 ±1.77	<0.001

Values are presented as mean ± standard deviation for continuous variables and *n* (%) for categorical variables; SD: standard deviation; CCVD: Cardio- and cerebrovascular disease; COPD: chronic obstructive pulmonary disease; Hs-CRP: high-sensitivity C-reactive protein; ^a^ Student’s *t*-test for continuous variables and Chi-square test for categorical variables.

**Table 2 ijerph-19-11585-t002:** Summary statistics of the residential air pollution exposure (2-year average) and meteorological factors (2-day average), and corresponding Spearman correlation coefficients.

Pollutant (Unit)	Mean ± SD	Max	Q3	Median	Q1	Min	IQR	Spearman’s Correlation Coefficient						
								PM_10_	PM_2.5_	SO_2_	NO_2_	O_3_	Temp	RH
PM_10_ (μg/m^3^)	48.34 ± 4.25	79.07	51.62	48.14	45.67	29.54	5.95	1.00	0.59 *	0.17 *	0.50 *	−0.66 *	0.002	−0.03 *
PM_2.5_ (μg/m^3^)	25.20 ± 2.69	39.07	26.85	24.89	23.15	12.95	3.70		1.00	−0.10 *	−0.13 *	−0.09 *	−0.001	0.13 *
SO_2_ (ppb)	5.40 ± 2.07	26.03	5.84	4.82	4.04	1.65	1.80			1.00	0.62 *	−0.22 *	0.02 *	−0.16 *
NO_2_ (ppb)	23.46 ± 7.96	40.81	30.13	21.28	17.60	1.00	12.54				1.00	−0.78 *	0.05 *	−0.24 *
O_3_ (ppb)	25.86 ± 2.97	44.27	28.23	26.16	23.71	18.07	4.52					1.00	−0.04 *	0.14 *
Temp (°C)	16.73 ± 8.60	33.08	23.21	18.95	11.35	-18.40	11.86						1.00	0.31 *
RH (%)	70.76 ± 12.32	99.95	79.84	71.73	62.82	18.96	17.02							1.00

SD: standard deviation; Max: maximum; Min: minimum; Q1: first quartile; Q3: third quartile; IQR: interquartile range; PM_10_: particulate matter with aerodynamic diameter < 10 μm; PM_2.5_: particulate matter with aerodynamic diameter < 2.5 μm; SO_2_: sulfur dioxide; NO_2_: nitrogen dioxide; O_3_: ozone; Temp: temperature; RH: relative humidity; * *p* < 0.001.

## Data Availability

The data in this study were from the Korean Genome and Epidemiology Study (KoGES; 4851-302), Korea National Institute of Health, Korea Disease Control and Prevention Agency, Korea.

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
