# Peer review of "Long-Term Effects of Ambient Particulate and Gaseous Pollutants on Serum High-Sensitivity C-Reactive Protein Levels: A Cross-Sectional Study Using KoGES-HEXA Data"

_ijerph, 2022, doi:10.3390/ijerph191811585_

Round 1
Reviewer 1 Report
1. Excepted for ambient air pollutants, CRP maybe associated with many other factors and this has been reported also in Table 1. So how to control the confoundings is key to the effects of air pollutants on CRP. However, only univariate analyses were shown in the manuscripts.
2. Either multiple linear regression or logistic regression has colinearity rules. As shown in the results, PM10, PM2.5, SO2, NO2 were correlated significantly. So it's unreasonable to use these correlated pollutants as predictors in the same model. The sensitivity analyses are questionable.
3. As to the main model in the supplementary results, BMI maybe correlated with blood pressure... The main model are also questionable. In addition, multivariate analysis has to have a sounding strategy of variates selection.
Reviewer 2 Report
Dear authors, firstly I would like to congratulate you on your document. The paper is interesting, original, clear, and well written, and I have added some suggestions to improve the overall quality of the document:
In the abstract section, please consider including the information that the odds ratio reported is indicative that the exposure might be a risk factor for the condition, however, it does not reflect a strong association.
In the results section, table 1, I found it difficult to understand the value of 2,001 (67.8) related to age, or 950 (32.2). I strongly suggest you make it clear what the values indicate and the units of measurement for each variable. For clarity, where is it possible to see, in table 1, that 67% of the participants were female?
In figures 2 and 3, please consider having the legend for the figure before the abbreviation meaning description.
Reviewer 3 Report
This manuscript presents a study on associations of long-term air pollution exposure on serum hs-CRP, adopting data from a Korean study. Although the associations between air pollution and CRP have been investigated previously, this study provide an interesting topic on the multiple measurement and long-term exposure-response relationship in Asian population. However, some issues should be considered.
1. Please provide the methodology for exposure assessment in detail. Although the CMAQ provided R2 values 0.81 for PM10 and 0.64 for PM2.5, the values of other concentration of gaseous pollutants (SO2, NO2, and O3) was no found. What’s the impact of exposure error on health effect estimation?
2. A set of individual-level covariates were adjusted in the model. Blood levels of white blood cells, fibrinogen, triglycerides, and fasting blood glucose were positively associated with PMs, SO2, and NO2, and inversely associated with O3 (Page 12, Lines 366-369). Did you consider the confounding effects of some important clinical parameters, such as glucose, blood lipid, cholesterol and triglycerides?
3. A discussion on the potential impacts of independent effects of pollutants and whether the chronic effect was attenuated by the acute effect (Page 5, Lines 175-179) is suggested.
4. The effect modification of drinking status should be presented (Table S2).
Round 2
Reviewer 1 Report
1. It's widely accepted that PMs are highly associated with NO2 or SO2. It's not appropriate to put these ambient air pollutants in the same linear/logistic regression model. As to the sensitivity analysis, I suggest to use the results of table a in the author's response.
2. The short-term exposures might have strong confounding effects according to the sensitivity analysis. It's better to add interaction item to the main model.
3. The statistics of medical history in table 1 is questionable. Please check and fix the data.
